# Computer-Generated Holograms Application in Security Printing

**Evgenii Y. Zlokazov** [1], **Vasilii V. Kolyuchkin** [2,*], **Dmitrii S. Lushnikov** [2] and **Andrei V. Smirnov** [3]

1  Department of Laser Physics, National Research Nuclear University MEPhI, 31 Kashirskoe Shosse, 115409 Moscow, Russia; eyzlokazov@mephi.ru
2  Laser and Optoelectronic Systems Department, Bauman Moscow State Technical University, 5/1 2nd Baumanskaya St., 105005 Moscow, Russia; dmlu@bmstu.ru
3  JSC "RPC "Krypten", 141980 Dubna, Moscow Region, Russia; smirnov_av@krypten.ru
*  Correspondence: vkolyuchkin@bmstu.ru

**Abstract:** In the present article the application of computer-generated holograms in security printing was considered. The main subject was creation of informative memory marks based on CGH for automatic read out of the information embedded in this marks that can be used for identification, copyrighting, or other types of authentication control of security elements. The fundamental theoretical basics of computer-generated holograms and its applications were given. The proposed technical solutions, such as "Printed hologram", "Smart verification", and "Holomemory", were considered in detail. The announced solutions were presented in the schemes of interaction between manufacturers of security printing and its potential consumers—citizens and regulatory government agencies.

**Keywords:** security printing; holography; computer-generated hologram; rainbow hologram; diffraction grating; diffractive optically variable image devices; photopolymer security hologram





## 1. Introduction

Computer-generated holograms [1] are widely used in different fields of science and technology. They are most popular in holographic memory systems [2], augmented reality projectors [3], wavefront sensors [4], and images spatial filtering and pattern recognition systems [5], etc. One of the application areas of computer-generated holograms (CGH) is security printing. Security printing usually means the print area characterized by special print methods used individually or in combination with optically variable features such as holograms aimed at protection against duplication or falsification of goods or documents. Identification documents (passports, different documents certifying the holder's identity such as a driver's license), banknotes, bank cards, excise stamps, and other valuable documents or controlled forms are often secured with such methods and features. Recently, along with physical security methods, digital methods such as QR codes, DataMatrix codes, and 1D and 2D codes, etc. have become widespread. In such cases, part of the appropriate data is decoded, and most often a connection to a database takes place on the basis of the decoded identifier to obtain the required information on the secured item.

CGH has been successfully used for a long time in applied optical technology, including but not limited to security holograms, also known as diffractive optically variable image devices (DOVIDs), e.g., to generate a so-called hidden image. In such cases, a point or coherent source is needed to restore the image from the hologram surface. The image can be restored and become visible either to the naked eye or with the use of simple portable devices. On the other hand, CGHs have been efficiently implemented in optical memory systems, where image is 2D amplitude and/or phase-modulated data page specially designed for automatic read-out [2]. It was shown that with the use of a simple SLM-based projection system, a holographic element with the square size of about 1 mm$^2$ with data capacity up to 1 MB can be obtained [6].

The article writers offer several unconventional ways of using CGH-based holographic memory in the area of security printing through modern analogue-digital methods of their generation, and information read-out (restoration) with portable data-processing computer modules such as smartphones or other devices where automatic control or data acquisition is required. This article considered the following use of CGH: as a printed hologram applied to the surface of paper, cardboard, metallized PET film, and other substrates by thermal-transfer printing or laser ablation; as a smart identifier on the surface of a rainbow security hologram restored as an optical image of DataMatrix code in the point source moved to an infinite distance; and as a personalized micro-CGH (non-recurrent, unique for each hologram) located on a photopolymer security hologram. Different algorithms of calculation, coding, and generation of CGH were selected for each area of application, and a smartphone with preinstalled specialized SW was taken as a direct or coded data read-out device.

## 2. CGH Basics

The main subject of computer holography is CGH, a real-world diffractive element with the fringe pattern model being numerically calculated and realized by displaying with a spatial light modulator (SLM) or printed on any type of surface or volumetric media. In order to restore the object imbedded onto CGH, a coherent or partially coherent optical setup is required. The advantages of computer holography are the absence of requirement of complex interferometric setup application for obtaining the real-world hologram and the possibility to record physically inexistent objects that are only presented as digital models.

Depending on the optical setup that is being used to read out the encoded object image, an appropriate model of hologram record setup must be simulated. The main problem in this process is representation (rendering) of the object wave complex amplitude function in the plane of the holographic carrier [1]. Using a model of the object, which can be represented as set of illuminating sources with amplitudes $\{a_n\}$, the object beam function can be found as the sum of elementary waves from each element of object model (Figure 1).

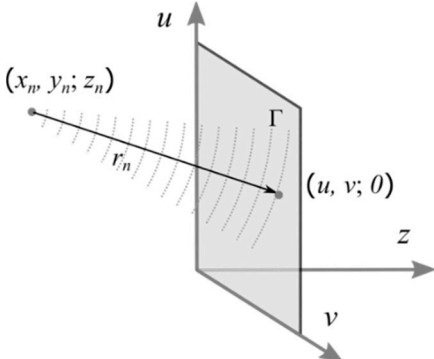

**Figure 1.** Schematic illustration of reference beam rendering process.

In the case of a point cloud model, each element illuminates spherical wave, and the object beam function can be found as (see also Figure 1):

$$A_{\text{ob}}(u_k, v_l) = \sum_{n=0}^{N-1} \frac{a_n}{r_{kln}} e^{j\left(\frac{2\pi}{\lambda} r_{kln} + \phi_n\right)}, \tag{1}$$

where $r_{kln}$ is the distance between illuminating point with coordinates $(x_n, y_n; z_n)$ to CGH point $(u_k, v_l; 0)$; $k = 0, \ldots K$ and $l = 0, \ldots L$ are integer numerators of object wave discrete model elements along $U$ and $V$ axes correspondingly, e.g.,

$$r_{kln} = \sqrt{(u_k - x_n)^2 + (v_l - y_n)^2 + z_n^2} \tag{2}$$

In the case of large values of $N$, $K$, and $L$, a direct calculation of Equation (1) requires significant computational burden. Different approaches are known to speed up this process. Look-up table methods use precalculation of spherical waves basic set and storing it in memory [7,8]. Those methods require a significant amount of memory usage and a lot of calculations must be performed. In the case of plane 2D objects, such as holographic memory data pages or with the use of planar slicing of 3D objects, model algorithms based on discretized forms of scalar diffraction theory integral transforms can be implemented in order to speed-up calculations. Depending on the propagation distance between CGH and image planes, discrete Fourier transformation for Fraunhofer diffraction zones (3a), discrete direct Fresnel transformation for far Fresnel zones (3b), or discrete angular spectrum propagation for short distances (3c) can be used in object beam rendering algorithms. These algorithms can be implemented in matrix form using the discrete fast Fourier transform (DFFT) operation as follows:

$$\mathbf{A}_{\mathrm{ob}} = \mathcal{F}[\mathbf{a}_{\mathrm{ob}}], \tag{3a}$$

$$\mathbf{A}_{\mathrm{ob}} = \mathcal{F}[\mathbf{a}_{\mathrm{ob}} \circ \mathbf{C}_{\mathrm{ob}}] \circ \mathbf{C}_{\mathrm{h}}, \tag{3b}$$

$$\mathbf{A}_{\mathrm{ob}} = \mathcal{F}\left[\mathcal{F}^{-1}[\mathbf{a}_{\mathrm{ob}}] \circ \mathbf{C}_{\mathrm{as}}\right], \tag{3c}$$

where $\mathcal{F}$—DFFT operation, $\mathcal{F}^{-1}$—inversed DFFT operation, and $\mathbf{C}_{\mathrm{ob}}$, $\mathbf{C}_{\mathrm{h}}$, and $\mathbf{C}_{\mathrm{as}}$ are spatial chirp matrices:

$$\mathbf{C}_{\mathrm{ob}} = \mathbf{C}_{\mathrm{as}} = \exp\left(j\pi\frac{\mu(m^2+n^2)}{N}\right); \mathbf{C}_{\mathrm{h}} = \exp\left(j\pi\frac{(m^2+n^2)}{\mu N}\right); \mu = \frac{\lambda z_0}{N\Delta_h}$$

where $\lambda$ is wavelength; $z_0$ is propagation distance; $N$ is the total size of the calculation field in pixels; and $\Delta_h$ is the discretization step in the CGH plane.

The next step of the CGH application process is the synthesis of the CGH fringe pattern discrete model. Most of the known holographic carriers have limited modulation characteristics. Purely phase, purely amplitude, mostly amplitude, binary phase, and binary amplitude are among the most popular. Though the values of function (1) are independent complex values, they must be encoded onto the points of carrier modulation characteristics. In the case of phase carriers, only the values of wrapped phase argument can be kept:

$$\Phi(u_k, v_l) = \arg[A_{\mathrm{ob}}(u_k, v_l)], \tag{4}$$

where $\arg\{\cdot\}$—argument operator. CGH obtained in such a way are known as simple kinoforms. The drawback of this approach is high spatial noise and distortions in the restored image scene due to the loss of object beam amplitude information. Iterative methods based on Gerchberg-Saxton algorithms [9–11], direct search with random trajectory method [12], and optimal projection algorithms [13] provide the decisions of kinoform fringe pattern function optimization problems (4) that yield a rise of restored scene quality:

$$\Phi_{\mathrm{opt}}(u_k, v_l) = \underset{\Phi_j(u_k,v_l)\in\Omega}{\mathrm{argmin}} \left[\sigma\{a_j, a_0\}\right], \tag{5}$$

where $\underset{x\in B}{\mathrm{argmin}}[A(x)]$ is the minimization operator of value $A$ of argument $x$ within a bounded set B; $\sigma\{a_j, a_0\}$ is a similarity measure between spatial amplitudes $a_0$ that is the object field point cloud model amplitude function, and $a_j$ is the object amplitude function restored by kinoform $\Phi_j(u_k, v_l)$ obtained on $j$-th search iteration, and $\Omega$ is the search rule specific for each of the mentioned iterative methods. The basic limitation of kinoforms is the loss and distortion of the object wave complex amplitude phase argument. This causes significant problems when the formation of 3D scenes is required. Additionally, the implementation of this method requires significant computational burden as soon as

iterative forward and backward transformations of the object field within object space and the CGH plane are needed.

One of the most promising approaches to operatively realize complex beam functions using a carrier with modulation characteristics limited to amplitude-only or phase-only types is the implementation of the bipolar intensity method, which simulates the classical off-axis Leith-Upatnieks scheme [14]. In that case, a plane wave unit amplitude reference beam model is included into record-simulating algorithms and the CGH fringe pattern function can be found as:

$$\Gamma(u_k, v_l) = \text{Real}\left[ A_{\text{ob}}(u_k, v_l) \cdot e^{-j\frac{2\pi}{\lambda}[u_k \sin\alpha + v_l \sin\beta]} \right] + B \tag{6}$$

where $\alpha$ and $\beta$ are reference beam directional angles in the record scheme model, and $B$ is constant bias term, which in the case of amplitude carrier must be selected to make $\Gamma(u_k, v_l)$ non-negative. Bipolar intensity method implementation provides a high diffraction efficiency CGHs capable of restoring amplitude and complex-valued data pages in archival holographic data storage systems [2,6], and 3D scenes [1]. In the case of plane images, application of random and pseudorandom phase masks allow obtaining high diffraction efficiency of holograms. However, the addition of an off-axis reference beam widens the spatial frequency spectrum of fringe pattern, limits the space bandwidth product of discrete carriers, and produces conjugate images in the observation plane. Additionally, in the case when a phase-only carrier is used the presence of high diffraction orders, it causes distortions in the restored image.

High quality restoration of an object image without an appearance of a conjugate image can be performed with phase-only carriers and application of the double-phase coding technique [15,16]. In this case, discrete complex values of $A_{\text{ob}}(u_k, v_l)$ are encoded by two phase-only discrete values of the carrier.

$$\Gamma(u_k, v_{\hat{l}}) = \begin{cases} \arg[A_{\text{ob}}(u_k, v_l)] + \arccos\frac{|A_{\text{ob}}(u_k, v_l)|}{2}, & \hat{l} = 2l \\ \arg[A_{\text{ob}}(u_k, v_l)] - \arccos\frac{|A_{\text{ob}}(u_k, v_l)|}{2}, & \hat{l} = 2l + 1 \end{cases} \tag{7}$$

where $\hat{l} = 1, \ldots 2L$ is integer numerator of fringe pattern discrete phase-only function elements along axis V. The problem of this method implementation is the necessity of exact control of the carrier phase shift when realizing the elements of $\Gamma(u_k, v_{\hat{l}})$. Quantization must be provided within the phase modulation depth on $2\pi$ radians.

The final step of CGH implementation process is the realization of the fringe pattern model on the physical carrier. Numerous methods are being used nowadays to perform this step: optical projecting of SLM aperture [17], polygraphic film printing [5], plasma-chemical etching [18], photo-lithography, and electron beam lithography [19], etc.

## 3. Printed Hologram

One of the simple ways to realize CGH-based informative elements is its printing on film using so called Computer-to-Film technology (CtF) and even on paper using an office printer. The main drawback of this approach is the limitation to binary amplitude modulation. In ref. [20], film-printed CGHs were implemented for complex Fourier filter holographic realization in 4-f correlator. It was shown that different screening techniques such as stochastic screening or diamond screening can be applied for grayscale patterns realization and distribution of spatial noise caused by quantization error can be controlled. However, the spatial resolution of the fringe pattern increases proportionally to the square root of the grayscale levels number. Binarization of CGH fringe pattern model does not affect the spatial resolution, but it causes the rise of quantization error in the restored field.

In refs. [21,22], it was demonstrated that binarized CGHs that form hidden 3D images can be imprinted onto a grayscale image by the use of CtF technology. Each element of the image was presented by binary sub-hologram, which forms a single-perspective 2D view of a hidden 2D or 3D object. Halftone levels of image elements are encoded by the

selection of a threshold level of corresponding sub-hologram fringe pattern binarization algorithms. Restoration of hidden image can be performed with the use of coherent light sources such as laser diodes or even LEDs. The problem of film-printed CGHs implementation in security applications is the complexity of restoring hidden objects using the same smartphone as both a light source and image detector due to the transmissive nature of such an optical element.

In ref. [23], it was shown that amplitude Fourier off-axis CGHs, designed according Equations (3a) and (6), can be printed on paper using thermal-transfer printing equipment or even an office printer and used as protective visible marks, named "Holocode", on products. These marks can be captured by smartphone cameras and encoded data objects or text can be numerically decoded. No coherent optical setup is used to read out the element as soon as the fringe pattern is captured by camera and subjected to post-processing using the algorithms based on scalar diffraction theory. The object image is restored numerically, which makes this method similar to the digital holography method, where the fringe pattern of a hologram is captured by a digital camera in a coherent interferometric scheme on a first step and the image is numerically restored on a second. The only difference is that we capture the fringe pattern of the printed Fourier CGH.

A Holocode as a marking method has fundamental differences from the well-known and widely used methods of marking, such as QR codes or DataMatrix codes. Experiments with attempts to distort the Holocodes showed that even in the case of up to 70% of surface distortions, a hidden image can be restored and analyzed. Moreover, different encryption techniques can be implemented in order to protect the authenticity of such a mark [24].

With the selected encoding algorithm, a sufficient quality of encoded data read-out and high computational speed, comparable to the speed of streaming printing, are provided. As an example, Figure 2 shows the process of converting the original logo "K" into a CGH and the subsequent read-out of this logo when the CGH fringe pattern image is captured with a smartphone camera. In addition to higher reliability indicators, a Holocode may be used either with an internet and database connection or as an independent tool for read-out of the total volume of information.

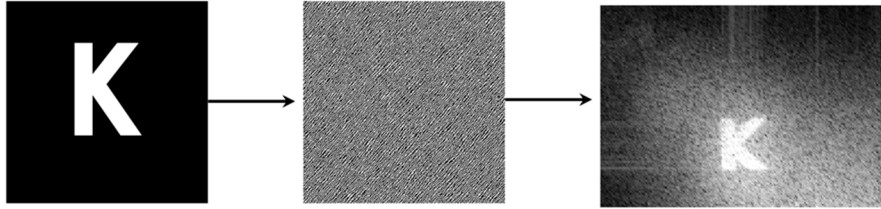

**Figure 2.** CGH Synthesis and read-out of the original logo "K".

The obtained method of Holocode generation and read-out may be used for marking any goods and may be applied to tamper-evident sealing labels or just to self-adhesive labels and contain manufacturer details, serial number, or any other official information. Holocode would allow the manufacturer and consumer to control the authentication of goods, or can be used as a marketing tool in case of database connection. The B2C interaction model with the use of Holocode is shown in Figure 3.

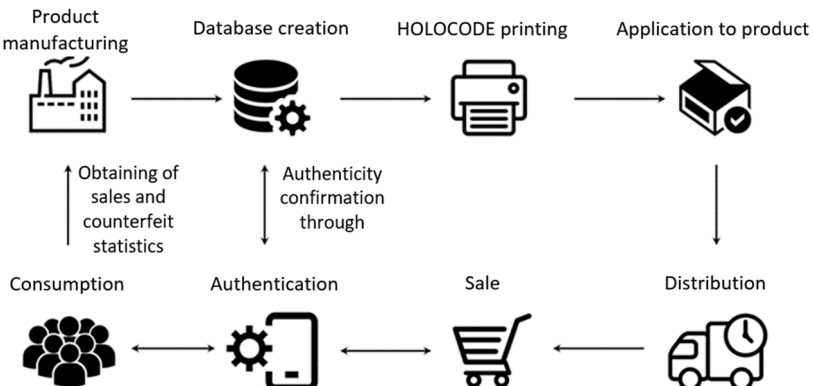

**Figure 3.** B2C interaction model with the use of Holocode.

## 4. Smart Verification

Another application of CGH is a hidden security feature combined with a classic rainbow hologram. In this case, a relief-phase hologram is supplemented with low-frequency relief, which a in zero-diffraction order forms a coded feature of a certain configuration observed and detected with a smartphone under specific conditions and while using a point source of visible light moved conventionally to an infinite distance. Such relief may be conventionally called pseudo-kinoform, though it deviates from the classic relief of this type due to a specific method of generation, while calculation is performed by discretization of the object light wave phase in the recording plane. As for a coded feature consisting of a set of points, kinoform was calculated according to Equation (4) or Equation (5). The kinoform recording was carried out using a frame-matrix holographer or electron beam lithography or optical lithography, as described in detail in the article [25], where a hidden image was realized by means of kinoform. Moreover, the image restored by laser radiation on the visualizer-screen did not contain an encoded data page, in contrast to the solution proposed by the authors.

A perfectly generated kinoform (Figure 4) restores only one image, either virtual or real, depending on what is used as a kinoform—a negative or a positive. For this purpose, it is required that the delay between the light falling on the area with 0 radian phase and the light falling on the area with $2\pi$ radian phase amounted to one wavelength exactly. In case of a reflecting kinoform, it means that the height of relief must be equal to half the wavelength or 270 nm for the green ($\lambda$ = 540 nm).

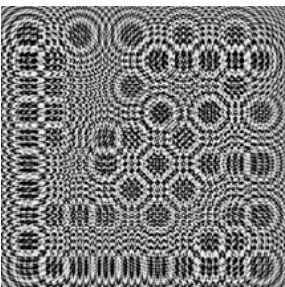

**Figure 4.** Synthesized kinoform fringe pattern.

The calculation of kinoform structures by the Gechberg-Sexton method was carried out in three cycles and 256 gradations of relief height were specified. When kinoform was placed on the diffraction grating, such deep relief of the kinoform worsened the quality of the main rainbow hologram. In order to avoid it, the depth of the kinoform relief must be several times less than the depth of the relief of the main hologram (Figure 5), while the kinoform itself will become similar to axial hologram where the real and virtual images are partially overlapping each other, and part of the light diffracts to the zero order, creating a bright spot in the center of the image. However, it is still possible to automatically read

out the DataMatrix image not overlapping with the main image and zero order. Thus, for the main hologram, the height of the relief was 300 nm with a spatial period of 1000 nm. For kinoforms, the height of the relief was about 100 nm with a spatial period of 5000 to 7000 nm.

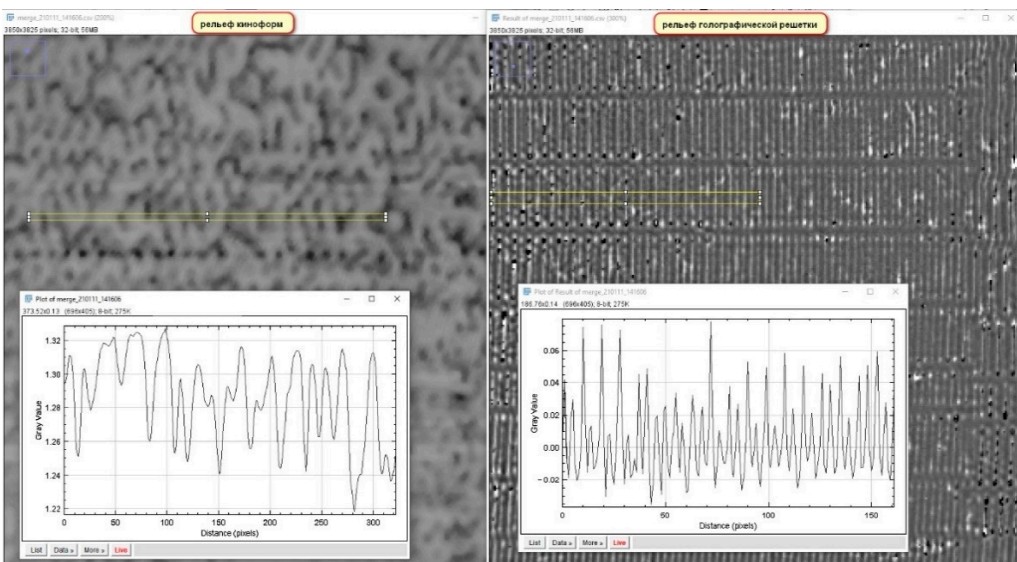

**Figure 5.** Profiles of kinoform and diffraction grating of the main hologram.

After making a hologram combined with kinoform relief by electron beam lithography or with the use of a frame-matrix holographer and having set the optimal exposure to restore both images (Figures 6 and 7), one can obtain an invisible hidden optical mark to be subsequently registered by a smartphone camera with LED flash.

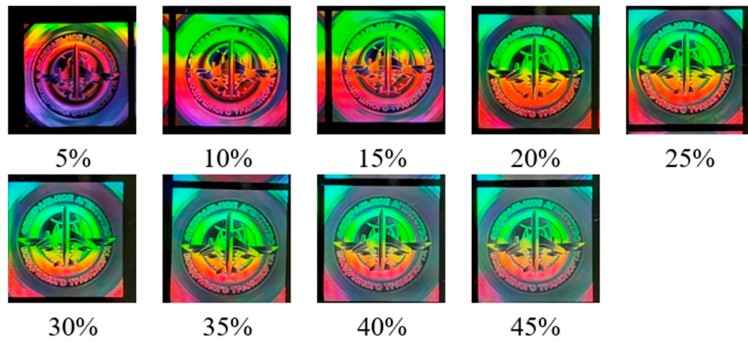

**Figure 6.** Main image of the hologram (exposure fitting).

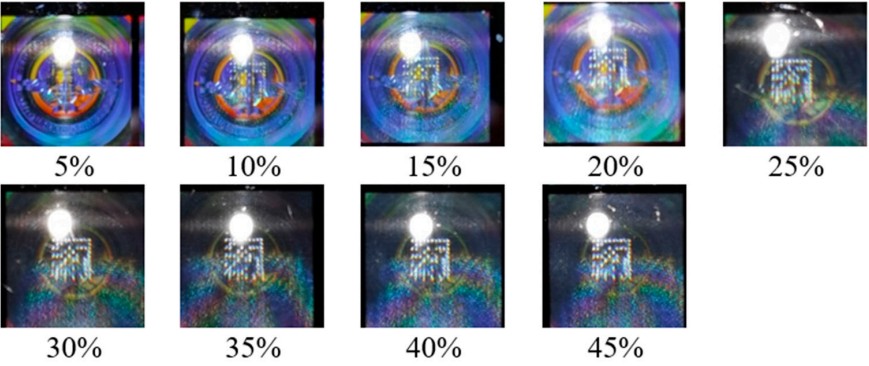

**Figure 7.** Hidden coded optical mark (exposure fitting).

To recognize the coded mark formed by the CGH, a common smartphone with a camera and LED backlight was used. Smartphone computing capacity was operated by a specialized Android-based SW designed for recognition and registration of marks. To ensure the focusing on the mark, a smooth change in the focal length in the range of 10–25 cm with increments of 0.5 cm was used based on an open smartphone camera interface.

For optical coded mark with CGH restored using an integrated LED of a smartphone, see Figure 8.

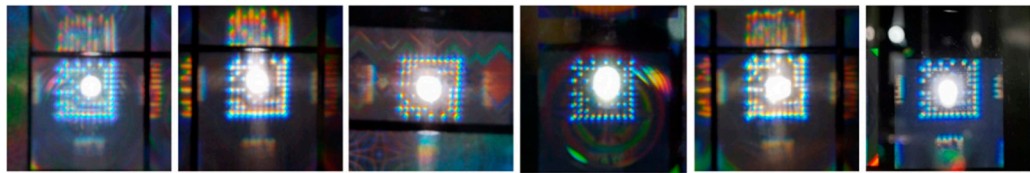

**Figure 8.** Restored optical coded mark with CGH using smartphone with LED flash.

If CGH is used as a coded mark, being an extra smartphone-readable feature in optically variable elements of security printing, it can be stated that protective functions are improved and another public feature that can be recognized by a user with a smartphone is added.

The model of interaction between the manufacturers of fast-moving consumer goods, security printing (holograms) providers, and consumers of secured goods using smart verification is shown in Figure 9.

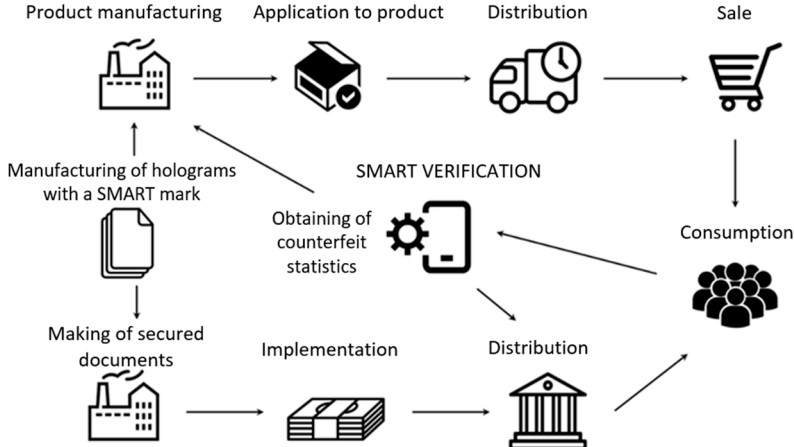

**Figure 9.** Model of B2C and B2B interaction with the use of smart verification.

## 5. Holomemory

Another alternative application of CGH is the offered method for photopolymer security holograms individualization [26]. The method consists of recording an extra computer-generated micro-hologram containing a hidden coded image with individual data in a small area of the photopolymer material.

The micro-CGH fringe pattern is synthesized and recorded as an extra sub-hologram in the defined area of the security hologram (SH). The micro-hologram is recorded by projecting a fringe pattern of computer-generated Fourier holograms that contain images with an individual digital code into the plane of photopolymer carrier. CGH is displayed in the spatial light modulator on the basis of liquid crystals.

When replicating the individualized SH, first the micro-CGH is exposed in accordance with the topology of the main holograms in the photopolymer strip, then the main 3D security hologram is recorded by placing it on the micro-CGH using the method of optical copying from master matrix. The main three-dimensional hologram is recorded by the Denisyuk method in the second exposure after recording the individualizing micro-CGH. The process of security hologram individualization must be carried out while continuously

moving the plate with the photopolymer carrier. At the same time, while the optical head transfers to the next SH, CGH with proper individual data for the *i*-th micro-hologram must be calculated. Therefore, every subsequent micro-hologram contains the required individualized information for every security hologram (Figure 10).

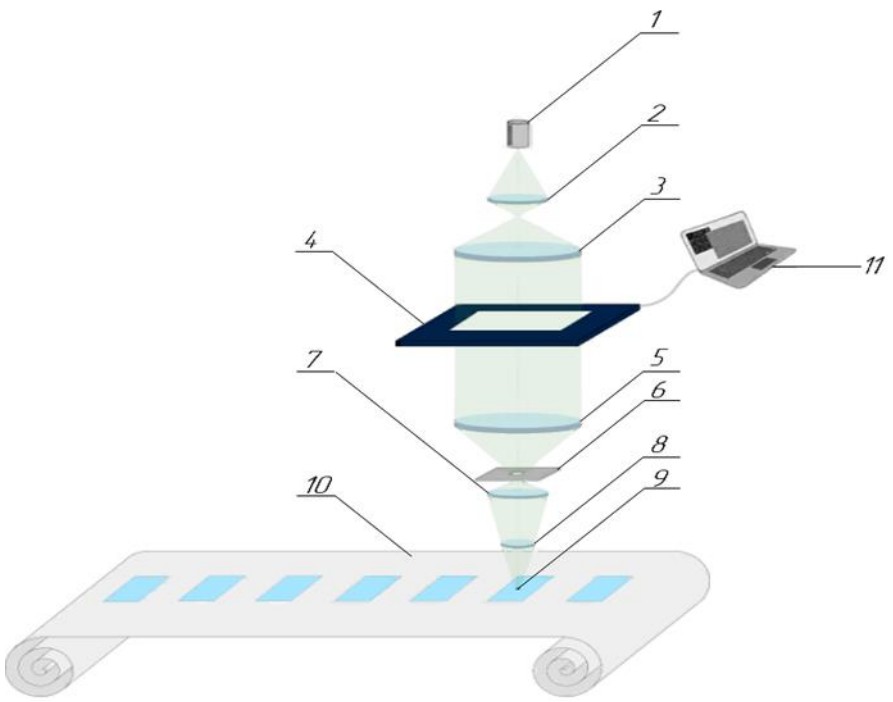

**Figure 10.** Photopolymer security holograms individualization: (**1**) is a LED or laser source; (**2**) is a condenser lens; (**3**) is a collimation lens of the lighting system; (**4**) is a spatial light modulator based on liquid crystals, where CGH is displayed; (**5**) is a Fourier lens; (**6**) is a spatial filter (diaphragm); (**7**) is a lens; (**8**) is a micro-lens; (**9**) is a personalized *i*-th micro-CGH; (**10**) is a photopolymer material; (**11**) is a personal computer for CGH calculation.

Using mathematical algorithms, the input digital information (in the form of digital texts, images, etc.) is coded with compression in the online mode and transformed into a coded image in the form of multidigit, e.g., 256-binary code. To generate a coded image, a coding algorithm from ECMA-378 [27] international standard which is used to record holographic memory disks (that is why the optical security element is called so) is applied. Additionally, the Cross Interleaved Reed-Solomon Code is used, which adds data redundancy and allows correction of damaged pixels of the coded image and complete restoration of data during subsequent read-out and decoding. Using the methods of computer-generation of Fourier holograms, the obtained coded image is transformed into computer-generated hologram presented in the form of gradation image (Figure 11). Algorithms based on Equations (3a) and (6) ensure their quick generation for seconds, while the data capacity of input information can be up to 1 Mb that is sufficient to store information in the form of digital identifier (digital code) consisting of a sequence number and link to the goods or documents from appropriate database where the data concerning the goods or documents have been entered.

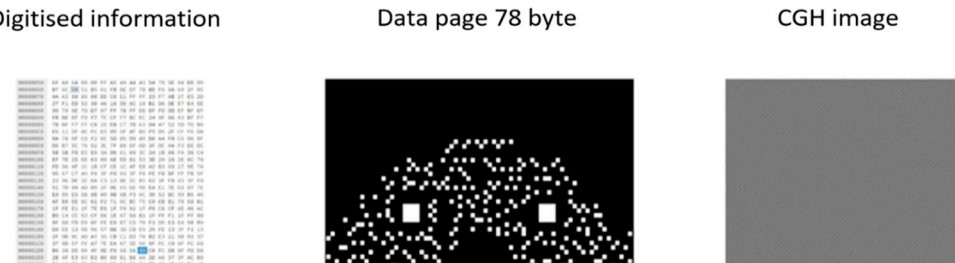

**Figure 11.** Stages of digital data transformation into CGH.

In the implemented scheme, the laser radiation with a wavelength of 532 nm illuminates the space light modulator with a converging light beam and transfers the image onto a photopolymer with a ten-fold reduction. The SLM dimension was $1024 \times 768$ elements and a pixel pitch was 36 μm. The resolution of the recorded micro-hologram was about $300 \ \text{mm}^{-1}$. The size of the recorded individualizing micro-hologram was 2 mm $\times$ 1.5 mm. To experimentally verify the recording method, text string KRYPTEN AB12345678 was taken as useful data to be recorded on the photopolymer material. Taking into account the redundancy introduced by means of the Reed–Solomon correction code, the input data were converted to 78 bytes. In our case, a region of $32 \times 32$ SLM elements corresponds to 1 byte in the data page after encoding. The generated data page had a dimension of 79,872 elements and corresponded to half the area of the SLM. To obtain a synthesized hologram that is uniform in illumination, a pseudo-random phase mask was added to the generated data page.

Using a projection optical system, sequential recording was performed. The result of consecutive recording of combined holograms will be an optical security element to be used to secure e.g., personal identification documents and containing level 1 visual features and micro-CGH which may be classified as expert level 3 feature requiring special devices for identification and reading. For combined exposure and enlarged image of recorded micro-CGH, see Figure 12.

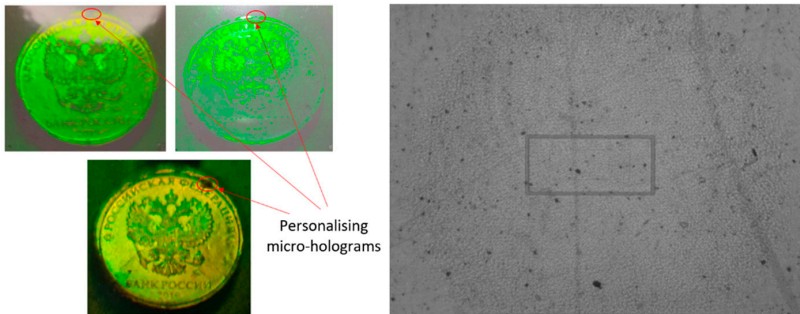

**Figure 12.** Examples of optical security element with micro-CGH.

The obtained micro-CGH may be read with an optoelectronic device, including a microscope type optical projection system with autofocus, LED light source and photodetector, and a protocol of data transfer to PC. The coded image of data page is restored on the read-out device by the use of fast inverse Fourier transform algorithm. An alternative read-out method is physical data recovery by illuminating the micro-hologram with coherent laser radiation using a Fourier transform objective and a matrix photodetector that registers the reconstructed encoded image [26]. In our case, preference was given to the optical projection reading method using incoherent radiation and digital restoration. This method was chosen to avoid the influence of a significant level of speckle noise from the hologram and the substrate to the reading result. Figure 13 shows the reconstructed encoded image, and its discrete fragments (pixels) are highlighted in green or red. Some data may be lost or damaged in the course of recording due to defocusing of the recording system or local defects of the carrier.

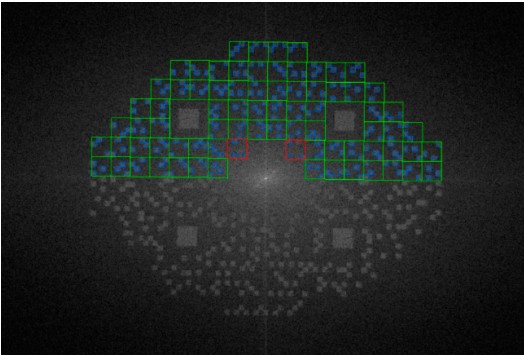

**Figure 13.** Reconstructed encoded image.

Correctly reconstructed pixels of the encoded image are highlighted in green color, and incorrectly reconstructed pixels in red. Therefore, inverse Reed-Solomon transform was used in the data decoding algorithms to recover data. In cases where at least 70% of the area was identified as correct, then the encoded image was corrected, and all data could be restored without loss. A screenshot of the program window with the result of reading the initial data, presented as the text string "KRYPTEN AB12345678", is shown in Figure 14.

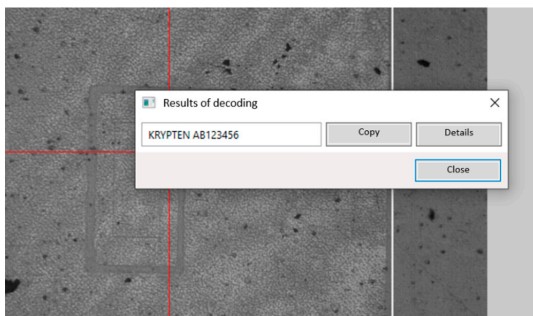

**Figure 14.** Result of successful decoding.

The developed method of obtaining and recording unique individualizing micro-CGH, used together with optical photopolymer security elements, may be applied to secure plastic documents in the form of embedded element, which guarantees impossibility to replace or falsify it without damaging the document integrity, as shown in the Figure 15.

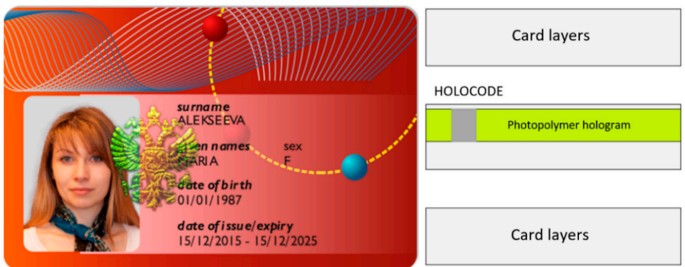

**Figure 15.** Example of an individualized micro-CGH.

The recorded individualized micro-holograms fulfil the functions required for document registering and commodity turnover, while the individual data cannot be read without a special device. As a result, the application of micro-CGH to obtain a secured material medium for individual data with modern data storage and exchange tools allows us to assume CGH application as a document security feature integrated into their creation, distribution, and control. For model of interaction between B2G and G2C with the use of Holomemory see Figure 16.

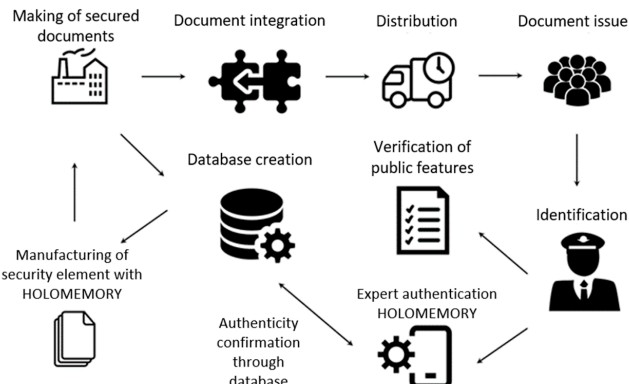

**Figure 16.** Model of interaction between B2G and G2C with the use of Holomemory.

## 6. Conclusions

In the present article, the described applications of holographic memory computer-generated holograms indicated a wide coverage of their use in security printing and informative marks production and management. The basic features of the CGH-based approach are simplicity of its realization that does not require complex interferometric setups, high repeatable accuracy, and the possibility of the usage of simple equipment suited with a digital camera, such as a smartphone, for originality verification. It can be confidently stated that the development of applications of computer-generated holograms remains relevant and promising. So, it is very likely that we will see many more interesting applications of computer-generated holograms in the future.

**Author Contributions:** Conceptualization, E.Y.Z. and V.V.K.; methodology, E.Y.Z., V.V.K. and D.S.L.; software, E.Y.Z.; validation, E.Y.Z., V.V.K., D.S.L. and A.V.S.; formal analysis, V.V.K.; investigation, V.V.K. and D.S.L.; resources, D.S.L. and A.V.S.; data curation, V.V.K.; writing—original draft preparation, E.Y.Z., V.V.K. and A.V.S.; writing—review and editing, E.Y.Z. and V.V.K.; visualization, D.S.L.; supervision, V.V.K.; project administration, V.V.K. and E.Y.Z.; funding acquisition, A.V.S. All authors have read and agreed to the published version of the manuscript.

**Funding:** This research was funded by JSC "RPC "KRYPTEN".

**Institutional Review Board Statement:** Not applicable.

**Informed Consent Statement:** Not applicable.

**Data Availability Statement:** Data are available from the authors upon request.

**Acknowledgments:** The authors thank all reviewers for their helpful comments and suggestions.

**Conflicts of Interest:** The authors declare no conflict of interest.

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
