# Peer review of "Computer-Generated Holograms Application in Security Printing"

_applsci, doi:10.3390/app12073289_

Round 1

Reviewer 1 Report

This manuscript provides a brief review on computer-generated holograms application in security printing. The overall content brings abundant information from the working CGH techniques, which might be interested by the related researchers. I have the following comments and suggestions:

  1. “Computer generated holograms” and “Computer-generated holograms” should be unified in the whole manuscript, including the title. Moreover, the abbreviation CGH was defined in Line 23, hence it could be directly used in Line 53.
  2. Point cloud model is introduced for calculating the object wave field as it is widely used in CGH algorithms. However, the propagation models used in iterative methods are normally not point cloud model. It is suggested to introduce other typical propagation models for generating CGHs, such as Fourier transform and Fresnel diffraction.
  3. The references about iterative methods are suggested to include the pioneer works of Prof. Wyrowski.
  4. AD is not defined in the text, which might be Aob defined in Eq. (1).

Author Response

Thanks for your constructive comments and suggestions!

Here are our answers point by point:

1) Thanks for noting. Fixed.

2) Technically, when plane image or data page is presented as digital image, each pixel of the image can be considered as point light source. So, scalar approximation integral transforms can be directly implemented to the whole image. We decided to add a brief representation of integral transforms algorithms in the text, see Eqs.3(a–c).

3) Thank you for the offer. We added the following link to references:

8. Frank Wyrowski and Olof Bryngdahl // Iterative Fourier-transform algorithm applied to computer holography / J. Opt. Soc. Am. A. 1988. Vol. 5, no. 7. P. 1058–1065.

4) Fixed.

Best regards!

Reviewer 2 Report

Authors define the manuscript entitled "Computer generated holograms application in security printing" as a review of fundamental theoretical basics of computer-generated holograms in the field of security printing. According to the authors security printing includes the prints made by special printing technologies used individually or in combination with optically variable features (i.e. holograms). The topic is interesting and relevant to the field because security elements on printed are aimed to protect the document against duplication or falsification.

Although the paper is promising, it is certainly not possible to include it in the review paper category. The authors present the applications of computer- generated holograms and does not belong to the group of review papers. For such categorization, detailed analyzes of existing research in this area are lacking, which should certainly be supplemented with published literature. Likewise, the paper requires an extensive review of the cited literature, which is presented sloppy, without a specific style.

If the authors want to keep the categorization of the paper, it is necessary to expand the literature and add other research in this area. Without such corrections, I do not recommend the paper for publication.

Author Response

Thanks for your comment. The area of CGH application in security printing is very diverse, so the significant expansion of the article is required in order to cover all the achievements in this field. We added the works by the leading groups in the field such as Ref.26 and 27 and compared the results presented in the revised version of the manuscript with the results described in these references. Initially main goal of the article is the revision of original results obtained by authors with short comparison with published results obtained by other researchers. If you decide that this is not enough for review category, it is worth to consider the possibility of changing the category of the article from review to regular paper. We will consider this case with the editor.

Reviewer 3 Report

Dear author,

Kindly go through my comments

Author Response

Thanks for your constructive comments and suggestions!

Here are our answers point by point:

1) The main subject of the article is creation of informative memory marks based on CGH for automatic read out of digital data embedded in these marks that can be used for identification, copyrighting or other types of authentication control of security elements. Such objects as data pages or DataMatrix codes, which can be captured and processed by portable devices such as smartphones, are considered as images for restoration by CGHs with the use of light sources like LEDs.

2) All the pictures in the article are the original results obtained by authors.

3) The area of CGH application in security printing is very diverse, so the significant expansion of the article is required in order to cover all the achievements in this field. We added the works by the leading groups in the field such as Ref.26 and 27 and compared the results presented in the revised version of the manuscript with the results described in these references. The main goal of the article is the revision of original results obtained by authors with short comparison with published results obtained by other researchers.

4) CtF method provides the holographic elements printed on film. Restoration of data is possible in this case if CGH is used as transparent element, which have to be illuminated by coherent beam. Holographic image is formed in a plane behind the CGH. This makes it impossible to use light source and digital camera from the same device, a smartphone for example. In the article we propose some other approaches such as printing CGHs on paper, realization of hidden images on rainbow security holograms and individualization of Denysiuk type security holograms.

5)

10.1515/lpts-2016-0036

In this article the method of security element with hidden image realization is based on combining the relief of original rainbow hologram with binary relief of CGH that restores hidden image using coherent read-out setup. No data page, DataMatrix or any other types of elements suitable for automatic read-out were used. We showed that in the case of Smart verification approach (Section 4) CGH can be realized in the similar way but saving all 256 levels of phase gradation. Also, DataMatrix image can be restored by smartphone LED and captured and automatically processed by the same smartphone. In the case of Holomemory approach, a main hologram is Denysiuk type hologram that were personalized by CGH.

6) 

https://doi.org/10.1364/AO.439004

In this case only CtF method is considered for realisation of main image with embedded hidden 3D image for visual observation. The only way for hidden image restoration is illumination of transparency type CGH by coherent beam. This method is hard to implement using light source and digital camera from the same device. In our manuscript we consider personification methods such as CGH printed on paper, CGH combined with rainbow hologram relief and CGH recorded above Denysyuk type hologram. We added the comparative analysis of holograms obtained in CtF machine with the holograms printed on film in Section 3 of the renewed version of manuscript.

7) 

https://doi.org/10.3390/polym13091358

In the given article the method of holographic element record is based on Denisyuk scheme. CGH is used to form an image of an object, which can be 3D image, QR code or ciphertext, in the plane of photopolymer-based holographic carrier. Factually the result is analogue thick hologram. In the cases shown in our paper CGH fringe patterns are directly realized on the carriers that are paper, film, rainbow hologram surface or photopolymer layer.

Also, in Section 4 and Section 5 we added some technical specificities of experimental implementation of CGHs that have been mentioned in the text of the manuscript in order to provide sufficient information for analysis of pros and cons of our methods.

Round 2

Reviewer 2 Report

The authors have made a certain progress in the manuscript and made the necessary corrections. I agree that the paper can be published. The editor should decide the categorization of the manuscript. 

Reviewer 3 Report

well done